

# Image classification adversarial attack with improved resizing transformation and ensemble models

Chenwei Li[1,2], Hengwei Zhang[1,2], Bo Yang[1,2] and Jindong Wang[1,2]

[1] State Key Laboratory of Mathematical Engineering and Advanced Computing, Zhengzhou, Henan, China
[2] Henan Key Laboratory of Information Security, Zhengzhou, Henan, China

## ABSTRACT

Convolutional neural networks have achieved great success in computer vision, but incorrect predictions would be output when applying intended perturbations on original input. These human-indistinguishable replicas are called adversarial examples, which on this feature can be used to evaluate network robustness and security. White-box attack success rate is considerable, when already knowing network structure and parameters. But in a black-box attack, the adversarial examples success rate is relatively low and the transferability remains to be improved. This article refers to model augmentation which is derived from data augmentation in training generalizable neural networks, and proposes resizing invariance method. The proposed method introduces improved resizing transformation to achieve model augmentation. In addition, ensemble models are used to generate more transferable adversarial examples. Extensive experiments verify the better performance of this method in comparison to other baseline methods including the original model augmentation method, and the black-box attack success rate is improved on both the normal models and defense models.

## INTRODUCTION

Convolution neural networks (CNNs) (*LeCun et al., 1989*) are widely used in image processing, such as image classification (*Krizhevsky, Sutskever & Hinton, 2017*), object detection (*Szegedy, Toshev & Erhan, 2013*) and semantic segmentation (*Milletari, Navab & Ahmadi, 2016*), most of which have better performance than human average capacity nowadays (*He et al., 2016*). Due to the better performance than other traditional deep neural networks, CNNs have derived a variety of network types to apply on different scenarios (*Alzubaidi et al., 2021*; *Naushad, Kaur & Ghaderpour, 2021*). However, when overlaying unnoticeable perturbations on original input, CNNs will mostly output incorrect predictions (*Szegedy et al., 2013*). This makes networks vulnerable to intended attacks and brings security problem to related models while CNNs have greatly facilitated our lives. These inputs added with specific perturbations are called adversarial examples. Misleading models to output incorrect result is firstly found in tradition machine learning (*Biggio et al., 2013*), and later in deep neural networks (*Szegedy et al., 2013*). Initial study

Corresponding author
Hengwei Zhang,
wlby_zzmy_henan@163.com

mostly focuses on image classification, and upcoming research find that adversarial examples also exist in other fields like real world (*Kurakin, Goodfellow & Bengio, 2018*). For instance, *Sharif et al. (2016)* demonstrated that the face recognition models are also possible to be evaded by physically realizable attacks. They made an adversarial eyeglass frame to impersonate an identity so that the face recognition model is fooled by an unauthorized person. While adversarial examples pose great threat to neural networks, it also can be used to evaluate the robustness of neural networks. Furthermore, adversarial examples can be used as extra training set to train more robust networks (*Madry et al., 2017*).

Existing methods of generating adversarial examples have already achieve considerable success rate in white-box attack, such as L-BFGS (*Szegedy et al., 2013*), C&W (*Carlini & Wagner, 2017*) and fast gradient sign methods (FGSM) (*Goodfellow, Shlens & Szegedy, 2014*). But the black-box attack success rate remains to be improved. FGSM provides a convenient way to generate adversarial examples with less computation and better transferability compared with other methods like L-BFGS and C&W. To further improve the black-box attack success rate, researchers derived more complicated methods from FGSM, referring to gradient optimization (*Zheng, Liu & Yin, 2021*), data augmentation (*Catak et al., 2021*) and ensemble models (*Chowdhury et al., 2021*), which are common methods to enhance the generalization of neural networks. Iterative version is firstly brought out like I-FGSM (*Kurakin, Goodfellow & Bengio, 2018*) and PGD (*Madry et al., 2017*). Momentum is then introduced to FGSM to relieve gradient oscillation and accelerate gradient convergence speed (*Dong et al., 2018*). Later, image affine transformation is used to manually increase original input and reduce overfitting in FGSM (*Xie et al., 2019*). Model ensemble is discussed to enhance adversarial examples performance in black-box condition (*Dong et al., 2018*; *Liu et al., 2016*). Table 1 shows the abbreviations used in this work.

The main contributions of this article are as follows:

This article refers to the idea of model augmentation and proposes the resizing invariance method (RIM) to enhance adversarial examples transferability. The proposed resizing transformation is verified to be eligible of invariance property and model augmentation. RIM utilizes the improved resizing transformation in generating adversarial examples to further increase input diversity. Similar to other data/model augmentation methods, RIM is readily combined with gradient optimization methods. In this article, momentum is used as baseline gradient optimization.

RIM can generate adversarial examples with ensemble models to realize higher black-box attack success rate than single model. In this article, we integrate four normal models to generate more transferable adversarial examples with RIM and test on three defense models, due to the fact that defense models are more robust than normal models and can better evaluate black-box attack performance.

Experiments on ImageNet dataset indicate that RIM has better transferability than current benchmark methods on both normal and adversarial-trained models. Additionally, the proposed method can further improve the black-box attack success rate

**Table 1 Acronym table.**

| Abbreviation | Full name |
|---|---|
| CNN | Convolution Neural Network |
| L-BFGS | Limited-memory Broyden Fletcher Goldfarb Shanno |
| C&W | Carlini Wagner |
| PGD | Projected Gradient Descent |
| FGSM | Fast Gradient Sign Method |
| I-FGSM | Iterative Fast Gradient Sign Method |
| MI-FGSM | Momentum Iterative Fast Gradient Sign Method |
| DIM | Diverse Input Method |
| TIM | Translation Invariance Method |
| SIM | Scale Invariance Method |
| RIM | Resizing Invariance Method |
| Inc-v3 | Inception-v3 Model |
| Inc-v4 | Inception-v4 Model |
| IncRes-v2 | Inception-Resnet-v2 Model |
| Res-101 | Resnet-101-v2 Model |
| Inc-v3-ens3 | Adversarial-trained Inception-v3 Model with Two Pre-trained Models |
| Inc-v3-ens4 | Adversarial-trained Inception-v3 Model with Three Pre-trained Models |
| IncRes-v2-ens | Adversarial-trained Inception-Resnet-v2 Model with Two Pre-trained Models |

when generating adversarial examples with ensemble models. This method is supposed to help evaluate network robustness and build more secure applications.

## RELATED WORKS

Image classification models are found to be evaded and output incorrectly with a simple gradient-based algorithm (*Biggio et al., 2013*). But this study is conducted merely on traditional networks like support vector machines, decision trees and neural networks. As for deeper neural networks like CNNs, *Szegedy et al. (2013)* applied hardly noticeable perturbations on original images by maximizing the model's prediction error to force the model misclassify those images. Their method requires relatively large computation compared with *Biggio et al. (2013)* as the linear search is inquired during each iteration. To reduce computation cost, *Goodfellow, Shlens & Szegedy (2014)* proposed the fast gradient sign method (FGSM) with only a single step and gradient query. FGSM can generate adversarial examples much faster than former methods, and most upcoming methods are derived from that.

Adversarial training is one of the effective ways to mitigate adversarial attacks (*Bai et al., 2021*). An intuitive thought is to directly put generated adversarial examples into network training. *Xie et al. (2017)* utilized their own diversity input method (DIM) in generating adversarial examples to adversarial-trained models. *Madry et al. (2017)* applied projected gradient descend (PGD) of their own in an expanding training set. But manually expanding a training set requires stronger attack methods than FGSM and larger capacity

networks in case of overfitting and learning nothing meaningful. Other than training networks with adversarial examples, *Tramèr et al. (2017)* proposed ensemble adversarial training method, using multiple pre-trained models for an ensemble adversarial training model to improve model robustness. A more convenient way to mitigate adversarial attacks is to purify and expunge adversarial perturbations on the original input, requiring less computation than training a new network. *Liao et al. (2018)* introduced a denoiser to avoid adversarial noise. *Guo et al. (2017)* compared multiple image transformation methods like image quilting, cropping, total variance minimization, bit-depth reduction and compression. However, generally speaking, image transformation defense effect is not as good as adversarial training. Thus, this article adopts three adversarial-trained models to verify proposed method black-box attack performance.

The process of generating adversarial examples for transferable image attack can be described as follows. Let $x$ be an original image and $x \subseteq X$ be the original input set. $y$ is the corresponding true label and $y \subseteq Y$ is the label set relevant to original input set. $L(x, y)$ is the model loss function which is usually cross-entropy. $x^*$ is adversarial example generated from $x$, and $y^*$ is the corresponding label. The aim is to find $x^*$ within maximum perturbation $\varepsilon$ so that $y^* \neq y$. The $x^*$ and $x$ should meet $L_\infty$ norm bound, *i.e.*, $||x^* - x||_\infty \leq \varepsilon$, to ensure the adversarial perturbation is hardly noticed by human eyes. For text convenience, the following methods should all meet this term. Considering CNNs loss function characteristic, the problem above could be transformed a condition constrained optimization as:

$$\arg \max_{x^*} L(x^*, y), \text{s.t.} ||x^* - x||_\infty \leq \varepsilon. \tag{1}$$

*Szegedy et al. (2013)* used L-BFGS method to transform the problem above into:

$$\text{minimize } c||x^* - x||_2 + L(x^*, y), \text{s.t.} x^* \in [0, 1]^m. \tag{2}$$

The purpose is to find minimum $c > 0$ with linear search and minimal $x^* - x$. The problem is transformed into a convex optimization to get an optimal approximate solution. Due to the linear search in every iteration, the computation is relatively large considering numerous parameters in both image and network. Later, *Carlini & Wagner (2017)* proposed a similar method named C&W to fulfill both targeted attack and non-targeted attack conditions. But their method has the same shortage as L-BFGS.

To reduce computation, *Goodfellow, Shlens & Szegedy (2014)* studied CNNs structure and decision boundary of clean images and adversarial examples. They utilized gradient information instead of global search, greatly reducing computation. The fast gradient sign method (FGSM) is proposed and proved to be equally effective in generating adversarial examples. The process is shown as:

$$x^* = x + \varepsilon \cdot sign(\nabla L(x, y)), \tag{3}$$

where *sign* is a function to decide perturbation direction. The white-box attack success rate is not ideal due to single step process, but this method is a benchmark that derives many improved versions.

*Kurakin, Goodfellow & Bengio (2018)* attempted to add perturbations in batches and proposed the iterative fast gradient sign method (I-FGSM). This modification substantially increased white-box attack success rate to almost 100%. The update equation is given as:

$$x_{n+1}^* = Clip_x^\varepsilon \{x_n^* + \alpha \cdot sign(\nabla L(x_n^*, y))\}, \tag{4}$$

in which $Clip_x^\varepsilon$ is to limit $x^*$ within $\varepsilon$ range of $x$, and $\alpha = \varepsilon/N$ is the perturbation in $N$ iteration. However, pure pursue on white-box attack success rate causes overfitting, leading massive decline in black-box attack success rate.

Referring to gradient optimization in training networks, *Dong et al. (2018)* introduced Momentum to generating adversarial examples and proposed the momentum iterative fast gradient sign method (MI-FGSM). Momentum could release overfitting to a certain degree, which can help escape poor local maximum and reduce gradient oscillation. By introducing momentum, the gradient information is memorized and used during every iteration. The update equation is shown as follows, in which $g_n$ is the accumulated momentum, and $\mu$ is momentum decay factor.

$$g_{n+1} = \mu \cdot g_n + \frac{\nabla L(x_n^*, y)}{||\nabla L(x_n^*, y)||_1}, \tag{5}$$

$$x_{n+1}^* = Clip_x^\varepsilon \{x_n^* + \alpha \cdot sign(g_{n+1})\}. \tag{6}$$

*Xie et al. (2019)* referred to data augmentation and proposed the diverse input method (DIM). DIM transforms input images with a certain probability once in an iteration and the transformation process is shown in Eq. (7). The transformation includes random resizing and random padding.

$$T(x_n^*, p) = \begin{cases} T(x_n^*), & \text{with probability p} \\ x_n^*, & \text{with probability } 1 - \text{p} \end{cases}, \tag{7}$$

$$g_{n+1} = \mu \cdot g_n + \frac{\nabla L(T(x_n^*, p), y)}{||\nabla L(T(x_n^*, p), y)||_1}. \tag{8}$$

To further improve transferability with data augmentation, *Dong et al. (2019)* proposed the translation invariance method (TIM) to enhance adversarial examples performance especially in black-box condition on defense models. Compared with original equation in Eqs. (1) and (9) shows the upgraded equation of TIM. $T_{ij}$ is the step that shifts image by $i$ pixels along up and down, and by $j$ pixels along left and right respectively.

$$\arg \max_{x^*} \sum_{i,j} w_{ij} L(T_{ij}(x^*), y), \text{s.t.} ||x^* - x||_\infty \le \varepsilon. \tag{9}$$

Instead of directly shifting gradient in TIM, *Lin et al. (2019)* took a step further of work *Xie et al. (2019)* and proposed the scale invariance method (SIM). Similar to DIM, SIM transforms images on pixel level. But SIM refers to scale transformation and transforms

multiple times in each iteration to achieve model augmentation. The update method is shown as:

$$\arg\max_{x^*} \frac{1}{m} \sum_i L(S_i(x^*), y), \text{s.t.} ||x^* - x||_\infty \leq \varepsilon \tag{10}$$

where $S_i(x) = x/2^i$ denotes the transformation copy and $m$ is the number of copies in each iteration.

Additionally, those data/model augmentation methods mentioned above are all readily combined with gradient optimization methods. In their works, MI-FGSM is used to conduct comparison experiments.

## RESIZING INVARIANCE METHOD

Overfitting is one of the main reasons to restrict the generalization of neural networks. Data augmentation, gradient optimization, ensemble models and early stopping (*Ali & Ramachandran, 2022*) are common approaches to relieve overfitting. Similarly, we believe the low black-box attack success rate is also caused by adversarial examples overfitting, and those approaches can also be applied to improve transferability. Baseline methods introduced above are the realizations of those approaches. In addition, SIM proposed model augmentation, which is a derived version of data augmentation, and achieved better black-box performance. Figure 1 shows the example of adversarial attack on two typical image classification models. The original image and adversarial image are hardly distinguishable by human eyes, and two typical models can correctly classify the original image as 'Lion'. But the adversarial image is misclassified as 'Sheepdog' and 'Persian Cat'.

Considering SIM utilized unvarying scale transformation and many other image transformations can also be used in model augmentation, we propose RIM, using random resizing transformation to realize model augmentation and improve adversarial examples transferability. The outlook of RIM is shown in Fig. 2 and the image transformation function is $R(x_n^*, p)$, randomly enlarging or reducing image. The update equation of $R(x_n^*, p)$ is shown as:

$$R(x_n^*, p) = \begin{cases} IE(x_n^*), & \text{with probability p} \\ IR(x_n^*), & \text{with probability } 1 - \text{p} \end{cases}. \tag{11}$$

In particular, $IE(x_n^*)$ enlarges the image to $rnd \times rnd \times 3$ in which $rnd \in [299, 330)$ and the surroundings are randomly padded to $330 \times 330 \times 3$. $IR(x_n^*)$ reduces the image to $rnd \times rnd \times 3$ in which $rnd \in (279, 299]$ and the surrounding are randomly padded to $299 \times 299 \times 3$. The image is transformed $m$ times in each iteration and $w_i$ is the weight for each loss.

According to *Lin et al. (2019)*, a loss-preserving transformation $T$ should satisfy $L(T(x), y) \approx L(x, y)$ for any $x \subseteq X$, and the corresponding label of $T(x)$ should be the same as $x$ to achieve model augmentation. Besides scale transformation, we find that resizing transformation is also a loss-preserving transformation, which is empirically verified in 'Experiments'. The improved resizing transformation is deployed multiple times in each iteration to meet model augmentation requirements, and the transformation not

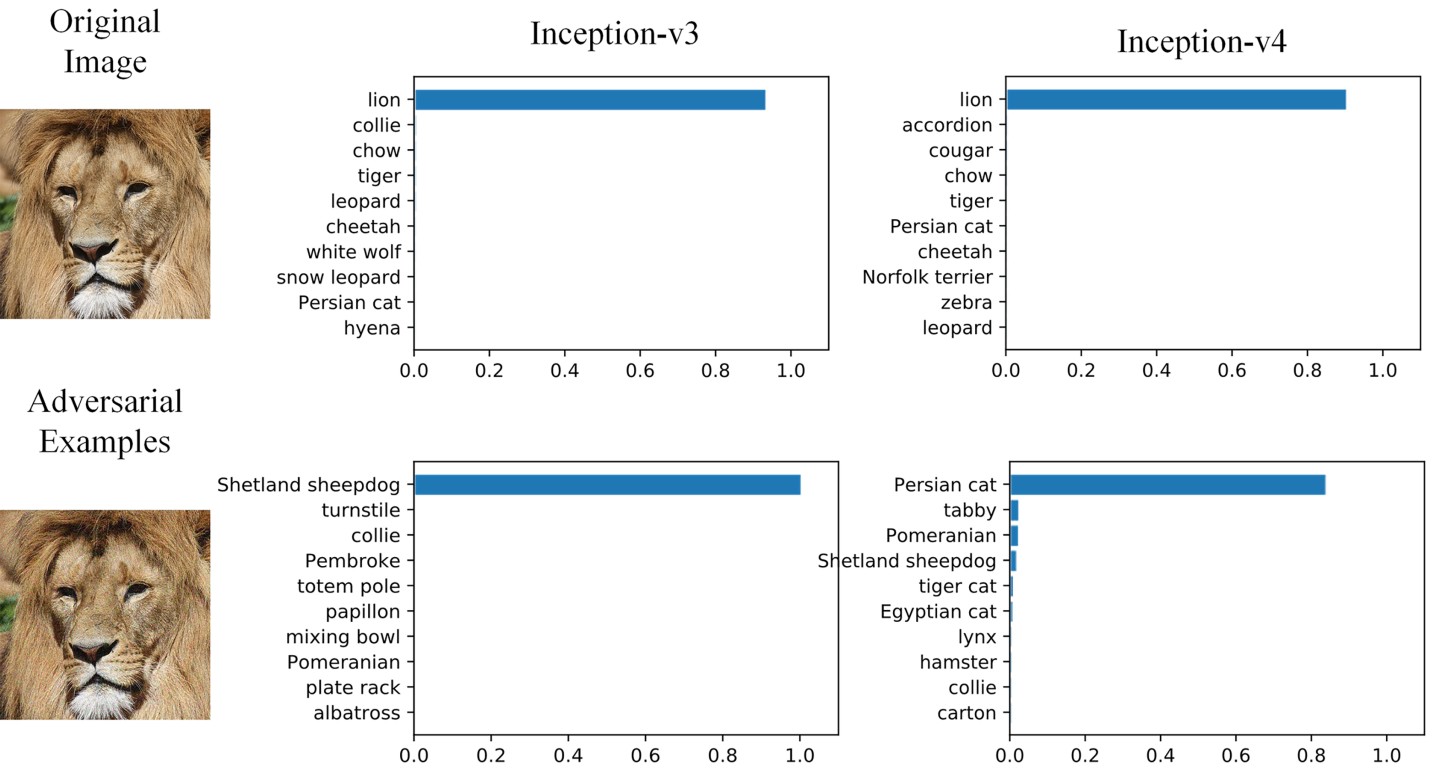

**Figure 1 Adversarial examples effect on image classification models.** Image source credit: Lion, https://www.hippopx.com/zh/lion-lion-s-mane-animals-90843, CC0, https://creativecommons.org/publicdomain/zero/1.0/deed.en.

only include image enlargement, but also include image reduction, which are the main differences from the resizing transformation in DIM.

## Single model generation

Using a known model to generate adversarial examples is a typical method to conduct image attack, transferable or not. In this article, we use four normal models as generating model and adversarial examples attack success rate are verified on both normal models and defense models. When verified on generating model, the condition is white-box, while on other models the condition is black-box. Similar to other data/model augmentation methods, RIM is readily combined with gradient optimization methods. Algorithm 1 shows the example that RIM combines with momentum for comparison convenience.

Due to the fact that white-box attack has already achieved considerable success rate, computation can be reduced and generating efficiency can be improved by not adopting our method but PGD or MI-FGSM. RIM can be degenerated to other methods by adjusting some parameters. For example, RIM degrades to DIM by setting $m = 1$ and $IR(x) = x$. Moreover, RIM degrades to MI-FGSM by setting $rnd = 299$.

Besides MI-FGSM, RIM can be combined with other gradient optimization methods such as Nesterov momentum or the adaptive gradient optimizer, only by replacing step 4 and step 5 in Algorithm 1 to their own update equations.

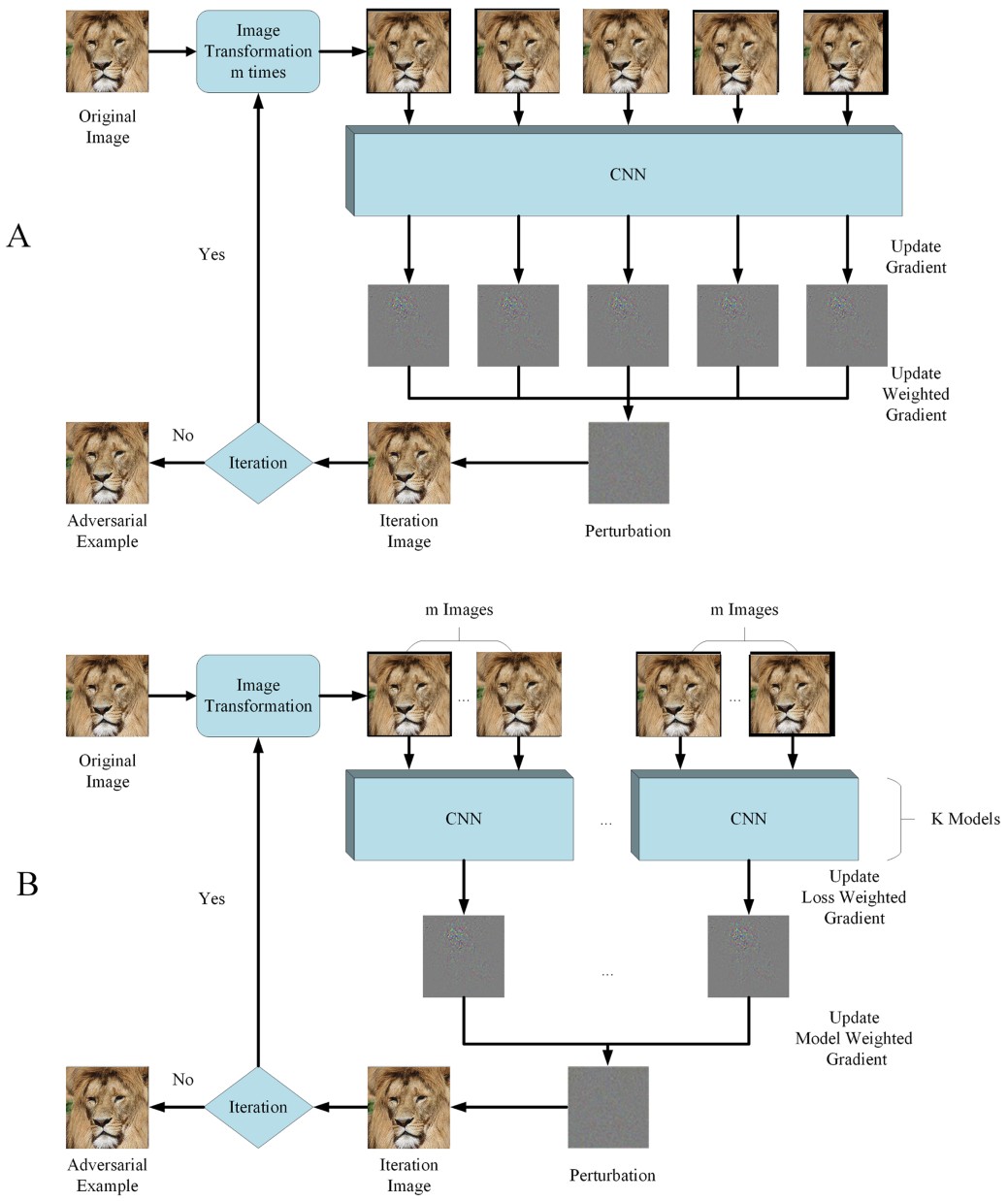

**Figure 2 Flowchart of RIM.** (A) With single model generation (B) with ensemble models generation. Image source credit: Lion, https://www.hippopx.com/zh/lion-lion-s-mane-animals-90843, CC0, https://creativecommons.org/publicdomain/zero/1.0/deed.en.

## Ensemble models generation

Model ensemble is a common way to train a model with more generalization. Several models are trained to vote output on test set. The generalization error is reduced by combining these models' output. Similarly, this method can be used on generating adversarial examples to improve transferability and reduce overfitting. Generating adversarial examples by ensemble models is different from single model. The original image is input to multiple models, from which getting different logits. These logits are weight-added as ensemble logits, and ensemble loss function is calculated with ensemble

---

**Algorithm 1  MI-RIM by single model.**

**Input:** original image $x$ and corresponding label $y$, maximum perturbation $\varepsilon$, iteration period $N$, momentum decay factor $\mu$, image transformation time in one iteration $m$, loss weight $w_i$, and transformation probability $p$.

**Output:** adversarial example $x^*$.

1. Initializing parameters: perturbation in each iteration $\alpha = \varepsilon/N$, $x_0^* = x$, $g_0 = 0$.

2. In iteration period $N$:

3. Update weighted loss function gradient $g = \sum_i w_i \nabla L(R_i(x_n^*, p), y)$,

4. Update accumulated loss function gradient $g_{n+1}$ by Eq. (5),

5. Update noise $x_{n+1}^*$ by Eq. (6),

6. Iteration ends, return $x^* = x_N^*$.

---

logits and the corresponding label. Logits are logarithmic to prediction for a network using cross-entropy loss function as Eq. (12) shows. $l(x)$ are logits and $1_y$ is one-hot code of label $y$.

$$L(x, y) = -1_y \cdot \log(\text{softmax}(l(x))). \tag{12}$$

Based on the work of *Dong et al. (2018)* and *Liu et al. (2016)*, the ensemble loss function is shown in Eq. (13). Suppose $K$ models are used to generate adversarial examples, and $w_k$ is the weight of each model.

$$L_e(x, y) = -\sum_k w_k 1_{yk} \cdot \log\left(\text{softmax}\left(\sum_k w_k l_k(x)\right)\right). \tag{13}$$

Similar to Algorithm 1, RIM by ensemble models' method is summarized in Algorithm 2, also combined with momentum for comparison convenience.

# EXPERIMENTS

Relatively comprehensive experiments are conducted to verify the effect of our method. Besides basic indicators, namely attack success rate, we consider different attack conditions and different parameters influencing adversarial examples performance. And the main purpose is to further enhance transferability and relieve overfitting while maintaining considerable white-box success rate.

One thousand images are randomly selected from ImageNet dataset (*Russakovsky et al., 2015*), belonging to 1,000 different categories. In other words, each image has a unique label. These images are able to be correctly classified by the models we use. After processed by the attack algorithm, 1,000 corresponding adversarial examples are generated. When the model misclassifies an adversarial example, we say a successful attack is achieved. And the higher attack success rate indicates better performance of the method. Preprocessing is applied to original input by transforming the format to PNG and the pixels are $299 \times 299 \times 3$ with RGB mode. Figure 3 shows the comparison of original images and adversarial examples. The difference is hardly noticeable by human eyes.

---

**Algorithm 2 MI-RIM by ensemble models.**

**Input:** original image $x$ and corresponding label $y$, maximum perturbation $\varepsilon$, iteration period $N$, momentum decay factor $\mu$, image transformation time in one iteration $m$, loss weight $w_i$, transformation probability $p$, number of models used to ensemble $K$, model weight $w_k$.

**Output:** adversarial example $x^*$.

1. Initializing parameters: perturbation in each iteration $\alpha = \varepsilon/N$, $x_0^* = x$, $g_0 = 0$.

2. In iteration period $N$:

3. Update $l_k(R_i(x_n^*, p))$ and $1_{yik}$ among $K$ models,

4. Update $L_e(R_i(x_n^*, p), y)$ by Eq. (13),

5. Update weighted loss function gradient $g = \sum_i w_i \nabla L_e(R_i(x_n^*, p), y)$,

6. Update accumulated loss function gradient $g_{n+1}$ by Eq. (5),

7. Update noise $x_{n+1}^*$ by Eq. (6),

8. Iteration ends, return $x^* = x_N^*$.

---

Seven models, *i.e.*, four normal models and three adversarial-trained models, are used in this article. Four normal models are Inception-v3 (Inc-v3) (*Szegedy et al., 2016*), Inception-v4 (Inc-v4) (*Szegedy et al., 2017*), Inception-Resnet-v2 (IncRes-v2) (*Szegedy et al., 2017*), and Resnet-v2-101 (Res-101) (*He et al., 2016*). Three defense models (*Tramèr et al., 2017*) are Inception-v3-adv-ens3 (Inc-v3-ens3), Inception-v3-adv-ens4 (Inc-v3-ens4), and Inception-Resnet-v2-adv-ens (IncRes-v2-ens). Generally speaking, defense models are more robust than normal models on adversarial attack, in both white-box and black-box conditions.

To prove the advantage of our method, we choose several baseline methods for comparison, such as MI-FGSM, DIM and TIM. The parameters are set as default according to these methods references. The maximum perturbation $\varepsilon = 16/255$ and the iteration period $N = 10$. For MI-FGSM, the momentum decay factor $\mu = 1$. For DIM, the transformation probability $p = 0.5$. For our method RIM, images are transformed $m = 5$ times in each iteration, and the loss weight $w_i = 1/m$. As for ensemble models, model weight $w_k = 1/K$.

## Invariant property

To verify that resizing transformation is loss-preserving and fits invariant property, we apply the proposed transformation on 1,000 images mentioned above with the resizing scale from 220 to 400 with a step size 10. Figure 4 shows the average loss of transformed images on three normal models, namely Inc-v3, IncRes-v2 and Res-101. We can see that the loss lines are relatively smooth and stable when the resizing scale is in range from 270 to 330, and the loss line on Res-101 oscillates severely outside that range. Thus, we presume that the resizing invariant property is in range from 270 to 330 among tested models.

As shown in Fig. 5, the accuracy of these transformed images keeps a relatively high level on three normal models. Especially, when the resizing range is from 260 to 340, the average accuracy is more than 96%. So, we utilize the proposed resizing transformation in

Original
Images

Resized
Images

Adversarial
Examples

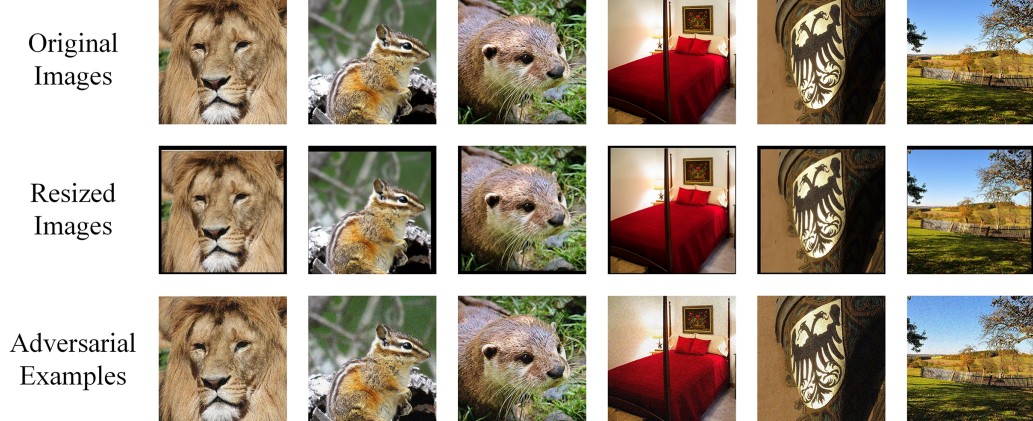

**Figure 3 Comparison of original images, RIM-transformed images and adversarial examples by RIM.** Image source credits: Lion, https://www.hippopx.com/zh/lion-lion-s-mane-animals-90843, CC0, https://creativecommons.org/publicdomain/zero/1.0/deed.en. Chipmunk: https://www.hippopx.com/zh/chipmunk-wildlife-nature-cute-420326; CC0, https://creativecommons.org/publicdomain/zero/1.0/deed.en. Otter: https://www.hippopx.com/zh/otter-otter-baby-otter-baby-nature-wildlife-photography-wild-animals-animal-155942; CC0, https://creativecommons.org/publicdomain/zero/1.0/deed.en. Bedroom: https://www.hippopx.com/zh/bedroom-guest-room-sleep-bed-chamber-boudoir-bedding-465007; CC0, https://creativecommons.org/publicdomain/zero/1.0/deed.en. Shield: https://www.hippopx.com/zh/coat-of-arms-adler-golden-shield-196592; CC0, https://creativecommons.org/publicdomain/zero/1.0/deed.en. Field: https://www.hippopx.com/zh/land-idyll-landscape-fence-agriculture-meadow-wood-fence-395426; CC0, https://creativecommons.org/publicdomain/zero/1.0/deed.en.

generating adversarial examples and achieve model augmentation to produce more transferable perturbations.

## Single model generation

We use baseline methods like MI-FGSM, DIM, TIM and SIM to compare the effect of RIM on boosting adversarial examples transferability and relieving overfitting. Adversarial examples are generated individually on four normal models with methods mentioned above, and tested on both four normal models and three defense models. Due to the inaccessibility to defense models, the black-box attack performance on defense models can better reflect transferability of these methods. Table 2 shows the experiments results on generating adversarial examples from single model. White-box condition is marked with *, and others are black-box conditions. Bold values indicate the highest success rate among these methods in single generating model.

White-box attack has already achieved considerable success rate. MI-FGSM does not adopt any data augmentation feature, but its white-box success rate still reaches more than 99%. Upcoming methods with data augmentation slightly descend on white-box success rate, but still maintaining relatively high level with at least 96%. RIM has average 98.4% success rate in white-box condition. In black-box condition, our method has better performance than baseline methods. For example, when generated from Res-101, RIM has average 67.3% success rate, 22.2% higher than second place SIM, which is only 45.1%. On more challenging defense models, RIM has average 41.1% success rate. Especially when

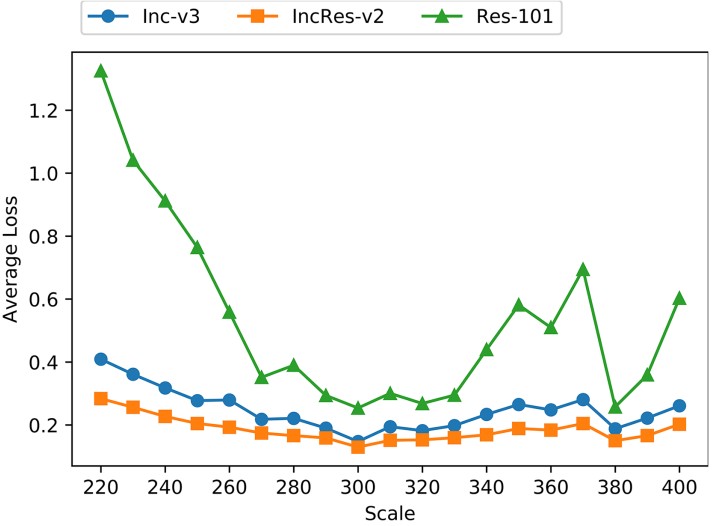

**Figure 4 Average loss of resizing transformed images on three models.**

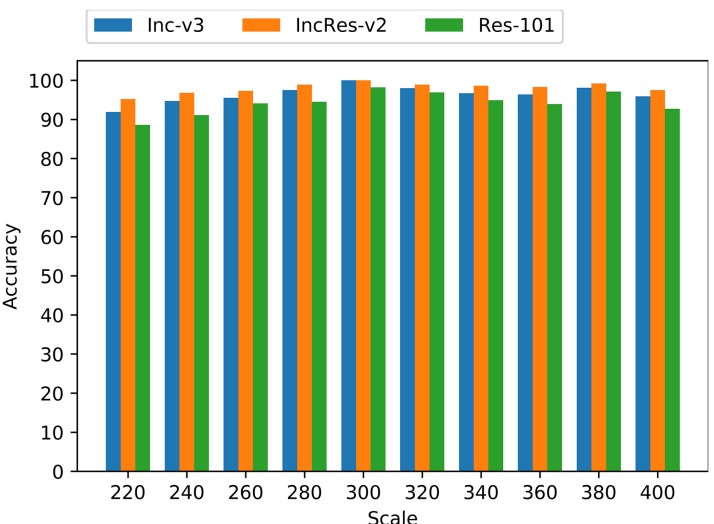

**Figure 5 Accuracy of resizing transformed images on three models.**

generated from IncRes-v2 and Res-101, RIM has more than 50% success rate on Inc-v3-ens3 and Inc-v3-ens4. However, RIM does not always outperform other baseline methods. TIM has better performance on IncRes-v2-ens when generated from IncRes-v2, and SIM has better performance on all three defense models when generated from Inc-v4. Thus, different methods can be adopted to fool a model, depending on whether the model is perfectly known or hardly known. Attack efficiency can be improved with proper method chosen. And if the model is absolutely unknown, we hope that our method can be first choice in black-box condition.

**Table 2 Success rate of adversarial examples with single model (unit: %).**

| Generating model | Method | Inc-v3 | Inc-v4 | IncRes-v2 | Res-101 | Inc-v3-ens3 | Inc-v3-ens4 | IncRes-v2-ens |
|---|---|---|---|---|---|---|---|---|
| Inc-v3 | MI-FGSM | 99.9* | 48.8 | 48.0 | 39.9 | 35.6 | 15.1 | 15.2 |
| | MI-DIM | 99.2* | 69.6 | 64.8 | 58.8 | 22.7 | 21.2 | 10.3 |
| | MI-TIM | 99.7* | 43.4 | 37.7 | 36.1 | 33.5 | 29.7 | 23.3 |
| | MI-SIM | **100.0***  | 70.9 | 68.7 | 65.5 | 33.1 | 31.5 | 17.3 |
| | MI-RIM | 98.4* | **87.3** | **82.9** | **77.7** | **38.3** | **36.1** | **17.5** |
| Inc-v4 | MI-FGSM | 65.6 | **99.9***  | 54.9 | 47.7 | 19.8 | 17.4 | 9.6 |
| | MI-DIM | 79.1 | 99.0* | 71.4 | 63.6 | 26.6 | 24.9 | 13.4 |
| | MI-TIM | 53.6 | 99.5* | 41.6 | 39.6 | 34.6 | 34.1 | 27.5 |
| | MI-SIM | 83.5 | 97.7* | 75.4 | 72.1 | **49.7** | **46.3** | **31.0** |
| | MI-RIM | **91.2** | 98.9* | **86.1** | **79.7** | 42.9 | 40.0 | 23.9 |
| IncRes-v2 | MI-FGSM | 69.8 | 62.1 | **99.5***  | 52.1 | 26.1 | 20.9 | 15.7 |
| | MI-DIM | 80.6 | 76.5 | 98.0* | 69.7 | 36.6 | 32.4 | 22.6 |
| | MI-TIM | 60.2 | 54.0 | 98.5* | 47.8 | 46.3 | 42.5 | **43.4** |
| | MI-SIM | 83.9 | 77.9 | 96.0* | 74.3 | 54.4 | 47.8 | 41.0 |
| | MI-RIM | **90.4** | **88.9** | 98.4* | **83.3** | **59.2** | **51.8** | 39.4 |
| Res-101 | MI-FGSM | 53.6 | 48.9 | 44.7 | **98.2***  | 22.1 | 21.7 | 12.9 |
| | MI-DIM | 71.0 | 65.1 | 62.6 | 97.5* | 32.4 | 29.8 | 17.9 |
| | MI-TIM | 42.3 | 36.6 | 34.3 | 98.0* | 37.6 | 33.3 | 28.7 |
| | MI-SIM | 66.4 | 59.9 | 57.4 | 97.0* | 34.7 | 32.2 | 20.0 |
| | MI-RIM | **89.2** | **84.5** | **85.6** | 97.9* | **56.5** | **51.7** | **36.2** |

Notes:
* White-box condition.
Bold values indicate the highest success rate among these methods in one generating model.

## Ensemble models generation

Suppose that three defense models are still unknown, we integrate four normal models as ensemble models. Adversarial examples are tested on both four normal models and three defense models. Thus, the condition is white-box when tested on four normal models individually, which is marked with asterisk (*) in Table 3.

Similar to single model generation, MI-FGSM holds better performance in white-box condition, while our method provides better transferability. Compared with single model generation, adversarial examples generated by ensemble models have better black-box attack success rate. For example, TIM has 67.2% average rate by ensemble models, which is nearly twice by its single model average. And SIM has 36.6% average rate by single model, almost half of 70.7% by ensemble models. On all three defense models, our method holds the best performance and proves to generate more transferable adversarial examples.

As shown in *Lin et al. (2019)* and *Yang et al. (2023)*, different data/model augmentation methods can also be combined to form a stronger method. *Lin et al. (2019)* combined three data augmentation methods (namely DIM, TIM and SIM in Table 3) with Nesterov momentum to achieve 93.5% average rate on defense models by ensemble models. *Yang et al. (2023)* extended this method by introducing another data augmentation method and an adaptive gradient optimizer. The average rate was enhanced to 95.3%. Given the fact

**Table 3 Success rate of adversarial examples with ensemble models (unit: %).**

| Method | Inc-v3 | Inc-v4 | IncRes-v2 | Res-101 | Inc-v3-ens3 | Inc-v3-ens4 | IncRes-v2-ens |
|---|---|---|---|---|---|---|---|
| MI-FGSM | **99.9***  | **99.7***  | **99.6***  | **99.8***  | 45.2 | 41.6 | 25.4 |
| MI-DIM | 99.4* | 98.7* | 97.9* | 97.9* | 63.1 | 57.8 | 39.2 |
| MI-TIM | 99.1* | 98.2* | 93.8* | 96.7* | 69.5 | 69.1 | 63.1 |
| MI-SIM | 98.7* | 98.2* | 97.6* | 98.4* | 78.6 | 72.4 | 61.1 |
| MI-RIM | 99.4* | 99.1* | 98.8* | 98.5* | **82.7** | 77 | **64.1** |

Notes:
* White-box condition.
Bold values indicate the highest success rate among these methods in one generating model.

that the data would be expanded twofold with each data augmentation method combined, the increase in black-box condition could be predictable. Thus, we mainly focus on single method performance in this article. The results above demonstrate the effect of our method in enhancing transferability and relieving overfitting of adversarial examples.

## Statistical analysis

Setting transformation probability $p$ in RIM is to control the diversity of model augmentation. We change the value of $p$ from 0 to 1 with 0.1 step size while keeping other parameters invariant and the effect is shown in Fig. 6. Figure 6A is the results of single model generation by Inc-v3 and Fig. 6B is the ensemble models generation. The generating model (Inc-v3) in Fig. 6A is randomly selected among four normal models that have similar point-fold lines trend, and the point-fold lines show similar trend with ensemble models generation, which means the transformation probability has similar effect on success rate of both single model generation and ensemble models generation. When $p = 0$ or $p = 1$, RIM degenerates to a constant model augmentation method. The white-box attack success rate hardly changes at each transformation probability, but black-box rate has an obvious drop when $p$ is 0 or 1. Transformation probability has little effect in single model generation when $p$ is other values than 0 or 1. And in ensemble models generation, the transferability also has little difference as long as the transformation is random. This shows the importance of stochastic process in data/model augmentation. Similar results can be found in *Xie et al. (2019)* and *Yang et al. (2022)*. Thus, to make sure the two kinds of transformation methods always come up in one iteration, we set $p = 0.5$ at last.

Image enlargement and reduction scale is respectively 330 and 270. We separately change one value while keeping another one invariant. Other parameters are default regarding to this section. Figure 7A is the results of image enlargement scale effect on success rate in single model generation by Inc-v4, and Fig. 7B is in ensemble models generation. The generating model (Inc-v4) in Fig. 7A is randomly selected among four normal models that have similar point-fold lines trend. We see that the first half of black-box attack line shows logarithmic growth, and the second half tends to be stable. The trend takes place at approximately 320 pixels. Thus, we choose 330 pixels as image enlargement scale for redundancy consideration. Figure 7C is the results of image reduction scale effect on success rate in single model by IncRes-v2, and Fig. 7D is in ensemble models generation. The generating model (IncRes-v2) in Fig. 7C is randomly selected among four

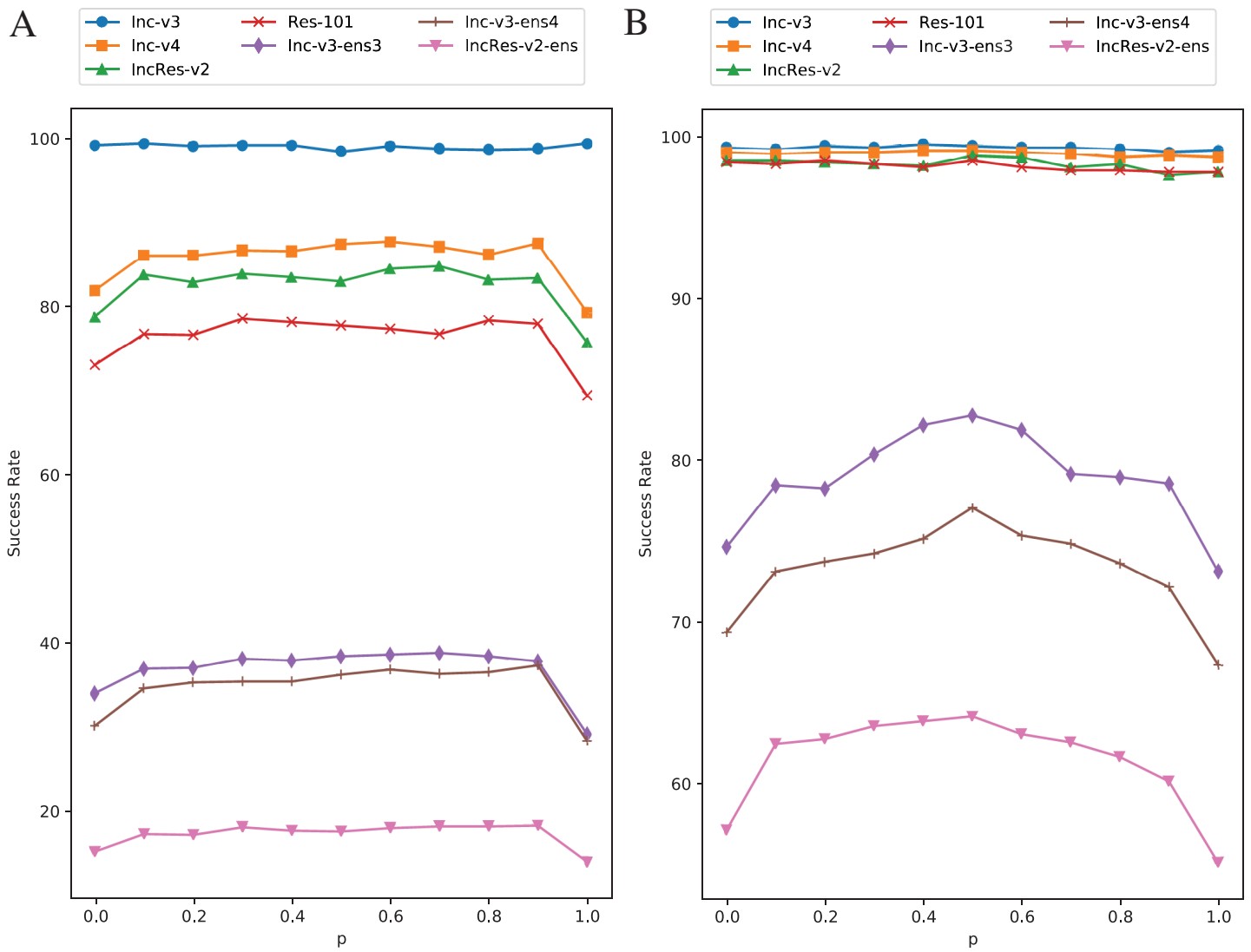

**Figure 6 Success rate of RIM on seven models at each transformation probability.** (A) Single model generation (Inc-v3). (B) Ensemble models generation. The generating model (Inc-v3) in (A) is randomly selected among four normal models that have similar point-fold lines trend.

normal models that have similar point-fold lines trend. The image reduction result shows similar feature as image enlargement. Especially, the logarithmic growth proves the benefits of resizing transformation in model augmentation.

Original images are transformed five times during each iteration. We increase the value of $m$ from one to ten with one step size to study the effect of resizing frequency. Figure 8 shows the relationship between attack success rate and transformation copies in each iteration. Figure 8A is in single model generation by Res-101 and Fig. 8B is in ensemble models generation. The generating model (Res-101) in Fig. 8A is randomly selected among four normal models that have similar point-fold lines trend. Black-box success rate gradually increases with more copies in each iteration. But more copies mean higher

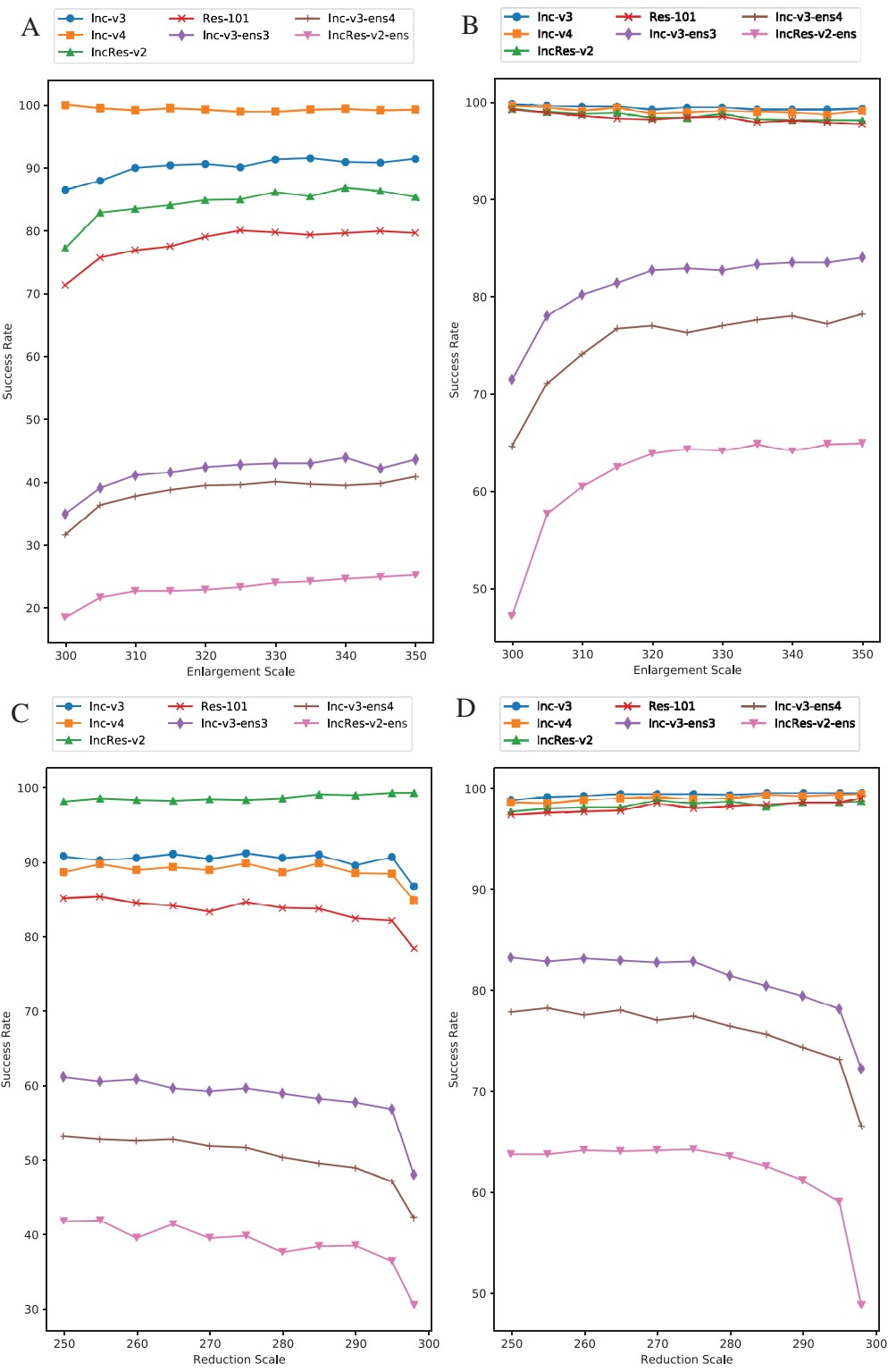

**Figure 7  Relationship between RIM success rate and image scale.** (A) Enlargement with single model generation (Inc-v4). (B) Enlargement with ensemble models generation. (C) Reduction with single model generation (IncRes-v2). (D) Reduction with ensemble models generation. The generating model (Inc-v4) in (A) is randomly selected among four normal models that have similar point-fold lines trend. The same goes for IncRes-v2 in (C).

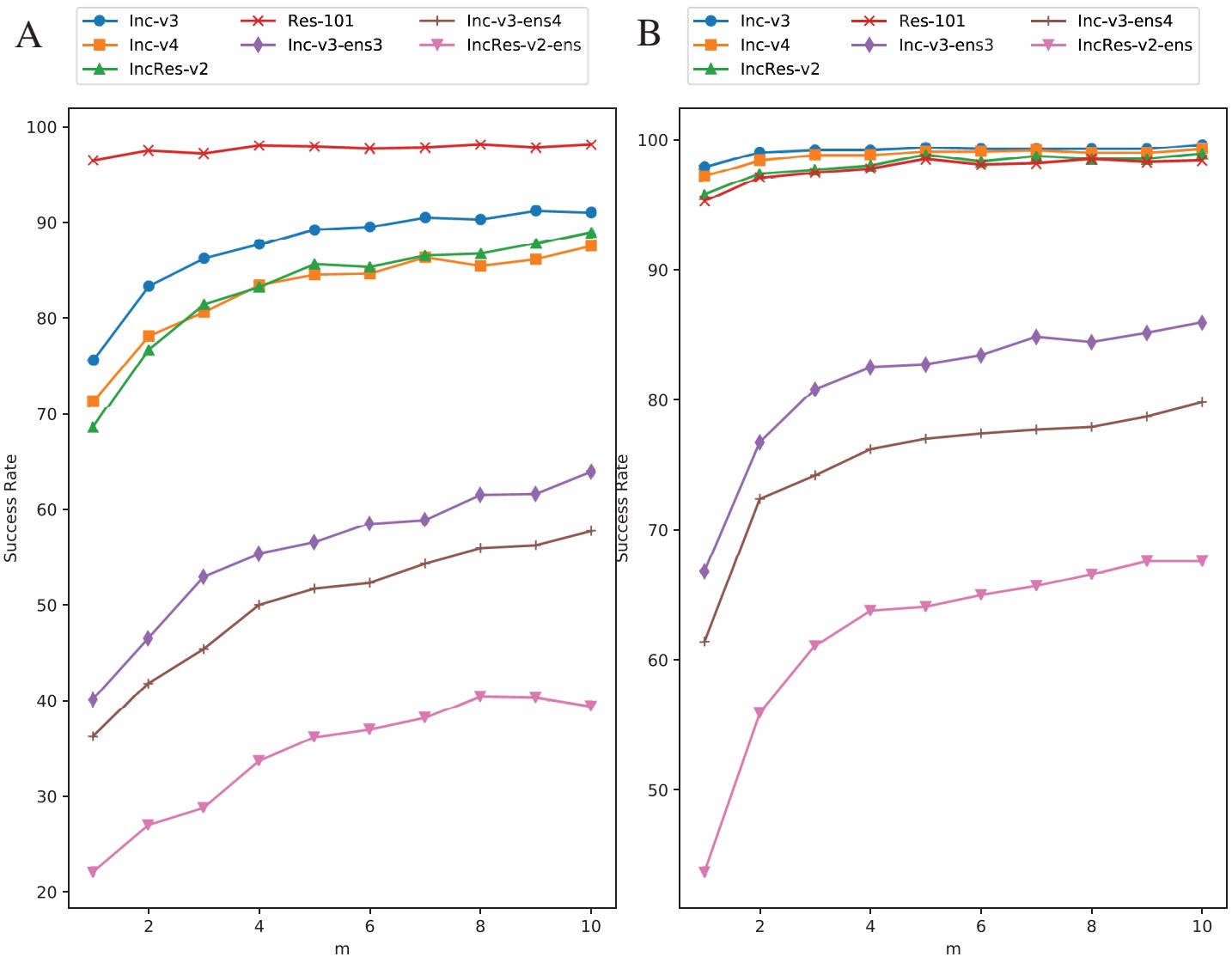

**Figure 8 Relationship between RIM success rate and number of transformed images.** (A) Single model generation (Res-101). (B) Ensemble models generation. The generating model (Res-101) in (A) is randomly selected among four normal models that have similar point-fold lines trend.

computation cost. To balance computation cost and black-box rate, we decide to set $m = 5$ considering the limitation of our experiment equipment.

## CONCLUSIONS

In this article, we propose the resizing invariance method in generating adversarial examples. RIM refers to data augmentation in training neural networks and introduces improved resizing transformation to achieve invariant property and model augmentation. Our main purpose is to improve adversarial examples transferability and relieve overfitting, which is directly reflected by black-box attack success rate. Ensemble models is introduced to further enhance transferability. Experiments are conducted on ImageNet

dataset to verify the effect of our method. Compared with baseline methods, RIM has higher black-box rate on normal models. As for more challenging defense models, RIM dominates the most models and has the highest average success rate. The advantage is more obvious in ensemble models with 74.6% average success rate. Our work proves the feasibility of model augmentation in image transferable attack, and other methods for enhancing network generalization are also likely to be used to enhance adversarial attack transferability.

Due to the multiple and random transformations in each iteration, RIM consumes more computation resources and takes longer computation time than other baseline methods. Also, although RIM holds better success rate on adversarial-trained models, it remains to be seen whether the adversarial examples generated by RIM can be directly used for adversarial training. We hope our attack method and models ensemble method can help develop more robust networks.

### Funding
This work was supported by the National Key Research and Development Program of China under Grant No. 2017YFB0801904. The funders had no role in study design, data collection and analysis, decision to publish, or preparation of the manuscript.

### Grant Disclosures
The following grant information was disclosed by the authors:
National Key Research and Development Program of China: 2017YFB0801904.

### Competing Interests
The authors declare that they have no competing interests.

### Author Contributions
- Chenwei Li conceived and designed the experiments, performed the experiments, analyzed the data, performed the computation work, prepared figures and/or tables, authored or reviewed drafts of the article, and approved the final draft.
- Hengwei Zhang conceived and designed the experiments, analyzed the data, prepared figures and/or tables, authored or reviewed drafts of the article, and approved the final draft.
- Bo Yang conceived and designed the experiments, performed the experiments, authored or reviewed drafts of the article, and approved the final draft.
- Jindong Wang conceived and designed the experiments, analyzed the data, prepared figures and/or tables, authored or reviewed drafts of the article, and approved the final draft.

### Data Availability
The ImageNet dataset is available at https://image-net.org/download.php. *Russakovsky et al. (2015)*. ImageNet Large Scale Visual Recognition Challenge. https://image-net.org/challenges/LSVRC/index.php.

The normal networks are available at GitHub and Zenodo:

- https://github.com/tensorflow/models/tree/master/research/slim.

- TensorFlow Developers. (2023). TensorFlow (v2.13.0-rc1). Zenodo. https://doi.org/10.5281/zenodo.7987192.

The defense networks are available at GitHub and Zenodo:

- https://github.com/tensorflow/models/tree/archive/research/adv_imagenet_models.

- TensorFlow Developers. (2023). TensorFlow (v2.13.0-rc1). Zenodo. https://doi.org/10.5281/zenodo.7987192.

The code is available at GitHub and Zenodo:

- https://github.com/NicAzrael/RIM.

- NicAzrael. (2023). NicAzrael/RIM: RIM (Tensorflow). Zenodo. https://doi.org/10.5281/zenodo.7979901.

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
