# Peer review of "Image classification adversarial attack with improved resizing transformation and ensemble models"

_PeerJ Computer Science, doi:10.7717/peerj-cs.1475_

## Round 0.1 · original submission · Minor Revisions

Dear authors,

Your paper has been reviewed. It requires minor revisions before being accepted for publication in this journal. I recommend that you improve the description of the contribution of your study, as one of the reviewers considers the contribution of your research to be weak.

Reviewer 1 ·

Basic reporting

Reviewer’s Report on the manuscript entitled:

Image classification adversarial attack with improved resizing transformation and ensemble models


The authors proposed a resizing invariance method to achieve model augmentation and through experiments show that the black-box attack success rate is improved compared to other baseline methods. The method and results are interesting, and the manuscript is well-written. Please see below my comments for further improvement.

Line 91. The related work can be further improved. The following articles can also be discussed and included:

The residual networks (ResNet) showed a good performance accuracy for image classification applications in land cover classification:
https://doi.org/10.3390/s21238083
Line 197. There are some techniques such as “early stopping” suggested in the article above that can prevent overfitting issues and reduce the computational cost. Please also mention it here.

Compressive domain deep CNN for image classification:
https://doi.org/10.3390/app12146881

Line 284. Please do not start a sentence with a number. Say “One thousand” instead of “1000”.

Line 352. Please use a better title here. You may say “Statistical analysis”.

Line 382. Please also mention the limitations of your method.

Figure 1. Some texts are overlapping “Shetland sheepdog”. Just say “Sheepdog”.

Please add an acronym table listing all the abbreviations used in this work.


Thank you for your contribution.

Regards,

Experimental design

Figure 4. Caption. Please add more descriptions for the graphs. In (A) you said (Inv-v3) and in (B) you said ensemble model. Did you mean these performed better. Please add a couple sentences in the caption of Figure 4. Similarly, for Figures 5 and 6.

Validity of the findings

No comment

Additional comments

Adding a flowchart showing the workflow of your research would help the general readers to follow your work easier.

Reviewer 2 ·

Basic reporting

The authors proposed an input diversity strategy coupled with iterative attacks. The aim is to overcome overfitting and improve the transferability of adversarial examples across networks trained for image classification. The input diversity is achieved by a transformation function randomly resizing and padding an image. This transformation is also considered when an ensemble of models generates adversarial examples.
The documentation of the paper is poor. The paper has many grammatical errors that need to be corrected.There is no continuity of the sentences with respect to the mathematical equations.Equation 9 and Equation 10 have shown the optimization problems without the constraints on the perturbation size.

Experimental design

The adversarial examples generated using a single model and an ensemble model scheme are transferred among similar normally and adversarially trained models. The proposed augmentation strategy shows a higher success rate than other iterative attack strategies with other model/data augmentation. Further, the authors perform ablation experiments to study the impact of the transformation
probability (p) and resize scale. The value of the transformation probability is varied p ϵ [0,1]. The transformation probability has a minimal effect on the success attack rate except at p = 0 and p = 1. The success rate increases when the image size is increased or reduced from its original size of 299x299x3.

Validity of the findings

The findings seem impressive. The authors should validate empirically if the transformation strategy mentioned in the paper is indeed an invariant property of deep neural networks.

Additional comments

This input diversity strategy presented in this paper is similar to the DIM method by Xie et al. The authors should highlight how their methods differ from DIM.

Reviewer 3 ·

Basic reporting

no comment

Experimental design

no comment

Validity of the findings

The article has low level impact although it is so long.

Additional comments

These article can be modified to be a survey. it is very log without any need to this long. The contribution of the article is weak. I did may best to extract the new in this article.

My advice for authors is to remove the unnecessary of sentences in the introduction section and also remove unelated works in the related work section. So, the reference section will be shrinked.

---

## Round 0.2 · accepted · Accept

Dear Authors,
Your paper has been accepted for publication in PeerJ Computer Science. I recommend carefully proofreading your manuscript before publication for typos/punctuation issues.

Reviewer 1 ·

Basic reporting

Dear authors,

Thank you for addressing my comments and improving your manuscript.

Regards,

Experimental design

no comment

Validity of the findings

no comment

Additional comments

Please carefully proofread your manuscript before publication for typos/punctuation issues.